# Vision-guided and Mask-enhanced Adaptive Denoising for Prompt-based Image Editing

## Abstract

Text-to-image diffusion models have demonstrated remarkable progress in synthe-sizing high-quality images from text prompts, which boosts researches on prompt-based image editing that edits a source image according to a target prompt. De-spite their advances, existing methods still encounter three key issues: 1) lim-ited capacity of the text prompt in guiding target image generation, 2) insufficient mining of word-to-patch and patch-to-patch relationships for grounding editing areas, and 3) unified editing strength for all regions during each denoising step. To address these issues, we present a Vision-guided and Mask-enhanced Adaptive Editing (ViMAEdit) method with three key novel designs. First, we propose to leverage image embeddings as explicit guidance to enhance the conventional tex-tual prompt-based denoising process, where a CLIP-based target image embed-ding estimation strategy is introduced. Second, we devise a self-attention-guided iterative editing area grounding strategy, which iteratively exploits patch-to-patch relationships conveyed by self-attention maps to refine those word-to-patch rela-tionships contained in cross-attention maps. Last, we present a spatially adaptive variance-guided sampling, which highlights sampling variances for critical image regions to promote the editing capability. Experimental results demonstrate the superior editing capacity of ViMAEdit over all existing methods.

## 1 Introduction

Image editing is a fundamental task in the field of computer graphics, with broad applications in var-ious areas such as gaming, animation, and advertising. In particular, text-guided image editing (Kim et al., 2022; Brooks et al., 2023; Kawar et al., 2023; Hertz et al., 2023), which allows images to be manipulated based on textual input, has gained significant attention due to its alignment with the way human communicates. Recently, driven by the powerful capacity of text-to-image diffusion models (Saharia et al., 2022; Ramesh et al., 2022; Rombach et al., 2022) for generating high-quality images from text prompt, prompt-based image editing (Hertz et al., 2023) has emerged as an effec-tive and efficient option for text-guided image editing. As illustrated in Figure 1a, prompt-based image editing aims to edit a source image according to a target prompt, which is derived by slightly modifying the given source prompt that describes the key content of the source image. The target prompt typically specifies the change of objects, their attributes, or the overall style of the image.

One prominent advantage of prompt-based image editing is that it enables training-free image edit-ing, where the pre-trained text-to-image diffusion model can be directly used for inference condi-tioned on the prompt without the need for expensive finetuning. Towards training-free prompt-based image editing, existing methods (Hertz et al., 2023; Mokady et al., 2023; Tumanyan et al., 2023; Cao et al., 2023; Parmar et al., 2023; Ju et al., 2024; Liu et al., 2024) typically adopt a dual-branch editing paradigm which involves a source branch and a target branch, both utilizing the same pre-trained text-to-image diffusion model as illustrated in Figure 1b. In this setup, the source branch works on reconstructing the source image conditioned on the source prompt, while the target branch aims at generating the desired target image guided by the target prompt. To ensure the structure consistency between the target and source images, both branches share the same noise variables throughout the diffusion denoising process. Although these methods have achieved remarkable progress, they still suffer from the following limitations.

Figure 1: Illustration of the prompt-based image editing task and the dual-branch editing paradigm.

- **L1: Limited capacity of the text prompt in guiding target image generation.** Existing methods generate the target image solely conditioned on the target prompt. Nevertheless, we argue that the target prompt can only highlight the core editing intention (*e.g.*, changing the given cat to a bear), whose capacity in depicting the finer visual details in the desired target image (*e.g.*, the color or expression of the bear, as shown in Figure 1) is inadequate. In fact, we can deduce a target image embedding from the source image embedding according to the editing intention conveyed by the source-target prompt pair. This deduced embedding can be explicitly integrated into the diffusion denoising process, providing a more precise guidance for generating the target image.

- **L2: Insufficient mining of word-to-patch and patch-to-patch relationships**. To preserve the background of the source image, existing methods mostly only rely on cross-attention maps which capture the word-to-patch relationships, to locate the editing area as patches relevant to the blend words[1]. Differently, the recent study, DPL (Yang et al., 2023), explores both word-to-patch and patch-to-patch relationships, where the patch-to-patch relationships reflected in self-attention maps are first utilized to cluster image patches into several groups, each corresponding to a candidate editing area, and then the cross-attention maps are used for determining the final editing area. Apparently, this method is inefficient due to the clustering operation, and lacks direct interactions between both kinds of relationships. In fact, the likelihood of a patch being relevant to the blend words is reflected in that of its related patches. Therefore, the patch-to-patch relationships can be used to refine the word-to-patch relationships towards more precise grounding of editing areas.

- **L3: Unified editing strength for all regions during each denoising step.** Existing methods apply the same sampling variance to all pixels during each denoising step, either using a fixed zero variance (Hertz et al., 2023) or a non-zero one (Huberman-Spiegelglas et al., 2024). However, we argue that the sampling variance controls the editing strength applied to each pixel, where the higher the sampling variance, the higher editing capacity it enables. Apparently, pixels in the critical image regions that are highly relevant to the target semantics (*e.g.*, the facial region of the cat in Figure 1) require stronger editing. Applying the same zero variance could result in insufficient edits to those critical regions, while using a uniform high variance for all pixels could distort the original image structure.

To address these limitations, we propose a Vision-guided and Mask-enhanced Adaptive Editing (Vi-MAEdit) method based on the dual branch editing paradigm, as illustrated in Figure 2. As the major novelty, unlike existing approaches that rely solely on the target prompt, our method explicitly exploits the given source image to guide the target image generation. In particular, we propose an image embedding-enhanced denoising process, where a simple yet effective CLIP-based target image embedding estimation strategy is introduced. This strategy aims to estimate the target image embedding by transforming the source image embedding according to the source and target prompt embeddings. In addition, we propose a self-attention-guided iterative editing area grounding strategy, which can leverage high-order patch-to-patch relationships conveyed by self-attention maps to refine the word-to-patch relationships, leading to more precise grounding results. Last but not least, for promoting the overall denoising process, we propose a spatially adaptive variance-guided sampling strategy, which applies higher sampling variance to critical image regions inside the editing area to promote the editing capability for these regions.

Overall, our contributions can be summarized as follows:

---

[1] Blend words refer to the words that specify attempted edits in the prompt, *e.g.*, "cat" and "bear" in Figure 1.

- We propose an image embedding-enhanced denoising process, where we present a target image embedding estimation strategy. Notably, we are the first to guide diffusion denoising process with additional image embeddings in the field of training-free prompt-based image editing.

- We introduce a self-attention-guided iterative editing area grounding strategy, which effectively exploits patch-to-patch relationships conveyed in self-attention maps as guidance to achieve more precise grounding of editing areas.

- We devise a spatially adaptive variance-guided sampling strategy, which enhances the model's editing capability to critical image regions, ensuring better alignment between the target image and prompt.

## 2 RELATED WORK

In this work, we focus on the task of training-free prompt-based image editing, in which both editing area grounding and diffusion sampling strategy play essential roles.

**Training-free Prompt-based Image Editing.** As a pioneer study, SDEdit (Meng et al., 2022) directly relies on a pre-trained text-to-image diffusion model to fulfil the task of training-free prompt-based image editing. This straightforward application of diffusion models makes it hard to preserve the essential content from the source image. To address this limitation, subsequent studies resorted to the dual-branch editing paradigm, comprising a source branch and a target branch as mentioned before. This paradigm enables the injection of specific features from the source branch into the target branch to control the target image generation. For example, P2P (Hertz et al., 2023) and PnP (Tumanyan et al., 2023) respectively inject cross-attention maps and self-attention maps from the source branch to the target branch to keep the structure consistency between the target image and the source image. Alternatively, MasaCtrl (Cao et al., 2023) injects key and value features in self-attention layers, to preserve the source image semantics for non-rigid editing, *e.g.*, pose transformation. Recently, InfEdit (Xu et al., 2024) presents a unified attention control mechanism that injects both attention maps and key-value features to boost both rigid and non-rigid editing. Despite their advances, these methods fail to fully exploit the visual cues during the denoising process, which is the major concern of our work.

**Editing Area Grounding for Image Editing.** To promote the image editing, early studies (Avrahami et al., 2022; 2023; Huang et al., 2023) employ a user-provided mask to indicate the editing area, and guide the target image generation. Nevertheless, the user-provided mask may be unavailable in practical applications. Accordingly, recent studies have focused on automatic editing area grounding. For example, DiffEdit (Couairon et al., 2023) grounds the editing area by comparing the noise predicted by the diffusion model respectively conditioned on the source prompt and target prompt. Later, P2P (Hertz et al., 2023) leverages the cross-attention maps computed by the diffusion model to locate the regions relevant to the blend words that specify the attempted edits. Recently, to fully utilize both cross-attention and self-attention maps, DPL (Yang et al., 2023) first clusters image patches into different groups using self-attention maps, and then selects groups corresponding to the editing area based on their average relevance to blend words according to cross-attention maps. Despite its great success, it overlooks the direct interactions between word-to-patch relationships captured by cross-attention maps and patch-to-patch relationships conveyed in self-attention maps. Therefore, beyond existing work, we propose a self-attention-guided iterative grounding strategy, which works on iteratively refining the word-to-patch relationships with patch-to-patch relationships.

**Diffusion Sampling for Image Editing**. Diffusion models generate real data by iteratively sampling cleaner data from previous noisy data, where the sampling strategy plays an important role. Currently, the mainstream sampling methods include DDIM sampling (Song et al., 2020) and DDPM sampling (Ho et al., 2020). Despite their compelling success, they typically need tens or hundreds of steps to generate high-quality real data. Therefore, many works have been dedicated to accelerating this process while maintaining the generation quality, *e.g.* using higher-order ODE-solvers (Lu et al., 2022a;b; Zheng et al., 2023; Zhao et al., 2024). Despite their advances in image generation, their capacity for image editing still needs to be explored. In this work, we found that the sampling variance significantly affects the editing strength to image regions. Motivated by this, we propose a spatially adaptive variance-guided sampling strategy, which enhances the editing strength to critical image regions, thus leading to better alignment with the editing intention.

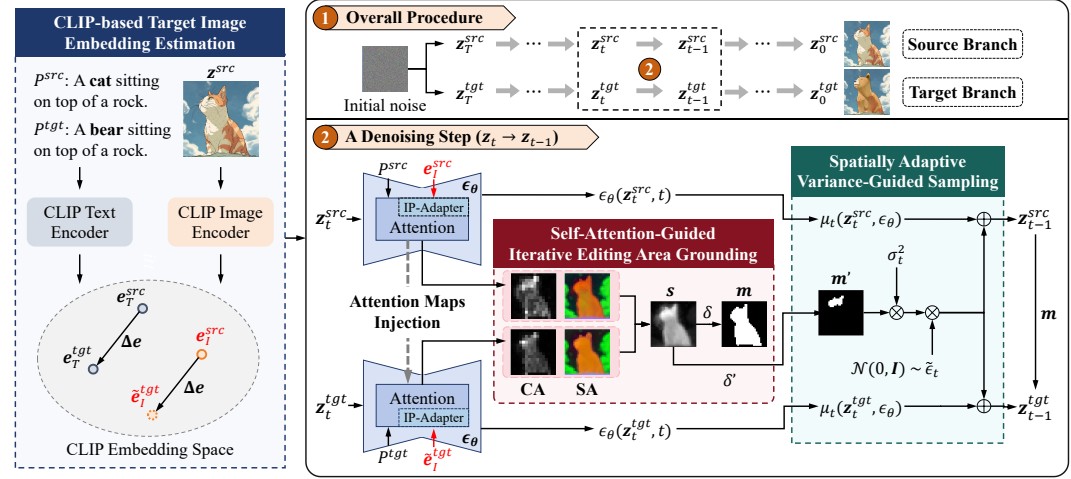

Figure 2: Overview of our proposed ViMAEdit for prompt-based image editing. CA and SA are short for cross-attention maps and self-attention maps, respectively.

## 3 PRELIMINARY

**Diffusion models** are a class of generative models, which generally involve two processes: forward process and backward process. The forward process transforms clean data from the prior data distribution to pure noise by iteratively adding random Gaussian noise, which is given by:

$$z_t = \sqrt{\alpha_t} z_0 + \sqrt{1 - \alpha_t} \epsilon_t, \quad t = 1, \ldots, T, \tag{1}$$

where $\epsilon_t \sim \mathcal{N}(0, \boldsymbol{I})$ denotes the noise added in the $t$-th timestep, and $\alpha_t$ controls the magnitude of the added noise. After $T$ timesteps, the resulting noise $z_T$ follows a standard Gaussian distribution.

Then the backward process targets to generate clean data by progressively removing the added noise from $z_T$. In each denoising step, cleaner data $z_{t-1}$ is derived by removing noise from previous data $z_t$. Typically, this is achieved by sampling cleaner data $z_{t-1}$ from a certain distribution, given by

$$z_{t-1} = \mu_t(z_t, \epsilon_\theta) + \sigma_t \tilde{\epsilon}_t, \quad t = T, \ldots, 1 \tag{2}$$

where $\tilde{\epsilon}_t$ is a random Gaussian noise, $\epsilon_\theta$ is a to-be-learned noise prediction model used for predicting the noise in noisy data $z_t$. $\mu_t(z_t, \epsilon_\theta)$ and $\sigma_t$ are the mean and variance of the distribution that $z_{t-1}$ can be sampled from. In the popular DDIM sampling (Song et al., 2020), they are defined as:

$$\begin{cases} \mu_t(z_t, \epsilon_\theta) = \sqrt{\alpha_{t-1}/\alpha_t}(z_t - \sqrt{1 - \alpha_t}\epsilon_\theta(z_t, t)) + \sqrt{1 - \alpha_{t-1} - \sigma_t^2}\epsilon_\theta(z_t, t), \\ \sigma_t = \mu\sqrt{(1 - \alpha_{t-1})/(1 - \alpha_t)}\sqrt{1 - \alpha_t/\alpha_{t-1}}, \end{cases} \tag{3}$$

where $\mu$ controls the magnitude of the variance. Once the noise prediction model $\epsilon_\theta$ is trained, new data can be generated according to Eqn. (2) by progressively denoising a random Gaussian noise.

Notably, the noise prediction model $\epsilon_\theta(z_t, t)$ is commonly implemented by a U-Net model (Ronneberger et al., 2015), which comprises a series of basic blocks. Each block mainly contains a residual layer and a self-attention layer. Specifically, the residual layer works on convolving the spatial features of the noisy data $z_t$ with a residual connection. In the self-attention layer, intermediate features obtained from the residual layer are projected into query features $\boldsymbol{Q}$, key features $\boldsymbol{K}$ and value features $\boldsymbol{V}$, and perform the self-attention mechanism given by:

$$\text{Attention}(\boldsymbol{Q}, \boldsymbol{K}, \boldsymbol{V}) = \boldsymbol{A}\boldsymbol{V}, \tag{4}$$

where $\boldsymbol{A} = \text{Softmax}(\boldsymbol{Q}\boldsymbol{K}^T/\sqrt{d})$ denotes the attention map where $d$ is the latent dimension.

## 4 METHOD

Formally, we denote the given source image, the source prompt, and the target prompt as $z^{src}$, $P^{src}$, and $P^{tgt}$, respectively. In this section, we will first introduce the basic dual-branch editing paradigm, and then present our proposed image embedding-enhanced denoising process, self-attention-guided iterative grounding strategy, and spatially adaptive variance-guided sampling strategy.

## 4.1 DUAL-BRANCH EDITING PARADIGM

As shown in Figure 1, the dual-branch editing paradigm commonly involves two diffusion model-based branches: a source branch for reconstructing the source image based on the source prompt , and a target branch for generating the target image guided by the target prompt. To ensure the structure consistency, both branches share the initial noise $z_T$ and all intermediate noise $\tilde{\epsilon}_t$.

Similar to the standard diffusion models, the text-to-image diffusion models used in the dual-branch editing paradigm also utilize the U-Net to implement the essential noise prediction model. Differently, they additionally input a text prompt. Accordingly, each basic block of the U-Net comprises not only the conventional residual layer and self-attention layer, but also a cross-attention layer to integrate prompt features. This cross-attention layer operates similarly to the self-attention layer, except that its key and value features are both projected from the textual features of the prompt, while the query features are still projected from the spatial features of the intermediate noisy data.

To guarantee high-quality editing results, the following two operations are commonly adopted.

- **Attention Maps Injection.** It has been observed that self- and cross-attention maps computed in the noise prediction model control the structure of the generated image. To preserve the source image structure, these attention maps in the target branch are replaced with those from the source branch. Formally, both self- and cross-attention layers in the target branch can be formulated as:

$$\text{Attention}(\boldsymbol{Q}^{src}, \boldsymbol{K}^{src}, \boldsymbol{V}^{tgt}) = \boldsymbol{A}^{src}\boldsymbol{V}^{tgt}, \tag{5}$$

  where $\boldsymbol{A}^{src}$ is the attention map calculated based on the query features $\boldsymbol{Q}^{src}$ and key features $\boldsymbol{K}^{src}$ from the source branch, and $\boldsymbol{V}^{tgt}$ is the value features from the target branch.

- **Editing Area Grounding.** To maintain the background (*i.e.*, the non-editing area) of the source image unchanged, the intermediate sample in the target branch at each denoising step would be further processed by replacing its non-editing area with that from the source branch, given by:

$$\hat{\boldsymbol{z}}_{t-1}^{tgt} = \boldsymbol{m} \odot \boldsymbol{z}_{t-1}^{tgt} + (1 - \boldsymbol{m}) \odot \boldsymbol{z}_{t-1}^{src}, \quad t = T, \dots, 1 \tag{6}$$

  where $\boldsymbol{m}$ denotes the binary mask of the grounded editing area. $\boldsymbol{z}_{t-1}^{src}$ and $\boldsymbol{z}_{t-1}^{tgt}$ respectively denote the original output noisy data in the source branch and the target branch at each denoising step, $\odot$ denotes the element-wise product. Ultimately, $\hat{\boldsymbol{z}}_{t-1}^{tgt}$ stands for the final processed noisy output at each timestep, which would be used as the input of the next denoising step.

## 4.2 IMAGE EMBEDDING-ENHANCED DENOISING PROCESS

Beyond existing methods that condition the denoising process of both branches solely on text prompts, we propose to incorporate the target image embedding as additional explicit guidance to promote the target image generation. Towards this end, we design a target image embedding estimation strategy based on embeddings of the source image and the source-target prompt pair.

Specifically, we resort to the CLIP (Radford et al., 2021) model, which embeds images and text into a unified embedding space with contrastive learning, and has shown powerful image and text embedding capability in various multimodal tasks (Shen et al., 2022). Let $\boldsymbol{e}_I^{src}$, $\boldsymbol{e}_T^{src}$, and $\boldsymbol{e}_T^{tgt}$ represent the source image embedding, the source prompt embedding, and the target prompt embedding extracted by the corresponding CLIP text and image encoder. We denote $\tilde{\boldsymbol{e}}_I^{tgt}$ as the estimated target image embedding. Considering that the difference between the target image embedding and the source image embedding is determined by the difference between the target prompt embedding and the source prompt embedding, we derive the target image embedding as follows:

$$\tilde{\boldsymbol{e}}_I^{tgt} = \boldsymbol{e}_I^{src} + (\boldsymbol{e}_T^{tgt} - \boldsymbol{e}_T^{src}). \tag{7}$$

Notably, to maintain the consistency between the source branch and the target branch, we also integrate the extracted source image embedding $\boldsymbol{e}_I^{src}$ into the source branch. Then, leveraging IP-Adapter (Ye et al., 2023), we feed the source/target image embedding alongside the source/target prompt into the noise prediction model $\epsilon_\theta$ in the source/target branch. During each denoising step of both branches, IP-Adapter first decomposes the obtained CLIP image embedding into a sequence of features via a linear projection layer. These image features are then integrated into the noise prediction model using additional cross-attention layers, which are formulated as:

$$\text{Attention}(\boldsymbol{Q}, \boldsymbol{K}', \boldsymbol{V}') = \lambda \boldsymbol{A}'V', \tag{8}$$

**Self-Attention-Guided Iterative Refinement**

Figure 3: Illustration of our self-attention-guided iterative editing area grounding strategy.

where $\boldsymbol{A}' = \mathrm{Softmax}(\boldsymbol{Q}\boldsymbol{K}'^T/\sqrt{d})$, $\boldsymbol{Q}$ is the query features of the intermediate noisy data, $\boldsymbol{K}'$ and $\boldsymbol{V}'$ denote the key and value features projected from the decomposed image features, respectively, and $\lambda$ is a pre-defined hyper-parameter to control the influence of the image embeddings.

## 4.3 SELF-ATTENTION-GUIDED ITERATIVE GROUNDING

As aforementioned, precise grounding of editing areas is essential for background preservation. Existing methods (Hertz et al., 2023; Xu et al., 2024) typically achieve this by simply using cross-attention maps between spatial features of the noisy data and token features of the text prompt. These cross-attention maps reveal the word-to-patch relationships, allowing for localizing the editing area as patches relevant to the blend words, *i.e.*, words that specify the intended edits. More recently, DPL (Yang et al., 2023) further uses self-attention maps, which capture the patch-to-patch relationships, to promote the editing area grounding. Specifically, it first clusters image patches into different groups using self-attention maps, and then determines their relevance to the editing area based on the average cross-attention scores of patches within each group. Despite its effectiveness, it overlooks the potential interactions between the word-to-patch relationships captured by cross-attention maps and the patch-to-patch relationships conveyed in self-attention maps.

We argue that the likelihood of a patch being relevant to a word is reflected in that of its related patches. Accordingly, we propose a self-attention-guided iterative grounding strategy, which works on leveraging the self-attention maps to iteratively refine the initial grounding result generated by the cross-attention maps. To derive the initial saliency map of the editing area, we follow existing methods (Hertz et al., 2023; Yang et al., 2023; Patashnik et al., 2023) to use the cross-attention maps yielded by the U-Net (defined in Eqn. (4)). Notably, within each U-Net block, each prompt word is associated with a cross-attention map that highlights its relevance to every image patch, while we only need the cross-attention maps that correspond to the blend word(s). Following previous studies, we average the cross-attention maps derived from multiple U-Net blocks with resolution $P_1$ ($P_1 = 16 \times 16$) to produce the initial saliency map, denoted as $\boldsymbol{s}_c \in \mathbb{R}^{P_1}$.

To improve the grounding result, we perform the self-attention-guided iterative refinement on the initial saliency map. In particular, we follow DPL to average self-attention maps from different U-Net blocks with resolution $P_2$ ($P_2 = 32 \times 32$) to construct the inter-patch relationship matrix, denoted as $\boldsymbol{Q} \in \mathbb{R}^{P_2 \times P_2}$. To refine the initial saliency map using this matrix, we first upsample the initial saliency map $\boldsymbol{s}_c$ to match the resolution of $\boldsymbol{Q}$, resulting in $\boldsymbol{s}_0 \in \mathbb{R}^{P_2}$. Then, to exploit the high-order relationships between different patches, we iteratively update the saliency of each patch by integrating the saliency of its related patches according to the inter-patch relationship matrix, which is formulated as:

$$\boldsymbol{s}_i = \gamma\boldsymbol{s}_{i-1} + (1 - \gamma)\boldsymbol{Q}\boldsymbol{s}_{i-1}, \quad i = 1, \ldots, N, \qquad (9)$$

where $\gamma$ is the trade-off hyper-parameter, and $N$ is the number of iterations. Ultimately, to facilitate the post-processing of the intermediate noisy data (see Eqn. (6)), we upsample the final saliency map $\boldsymbol{s}_N$, to keep its resolution as the same as that of the intermediate noisy data. Figure 3 summarizes the workflow of our self-attention-guided iterative editing area grounding strategy.

Let $\boldsymbol{s}^{src}$ and $\boldsymbol{s}^{tgt}$ be the final saliency map derived in the source branch and target branch, respectively. We follow existing methods (Hertz et al., 2023; Yang et al., 2023) to combine them to form the overall saliency map $\boldsymbol{s}$ of the entire editing area, which is given by:

$$\boldsymbol{s} = \max\left\{\boldsymbol{s}^{src}, \boldsymbol{s}^{tgt}\right\}. \qquad (10)$$

Then, we derive the binary mask $\boldsymbol{m}$ defined in Eqn. (6) with a pre-defined threshold $\delta$ over $\boldsymbol{s}$.

### 4.4 Spatially Adaptive Variance-Guided Sampling

As mentioned before, certain sampling strategy (*e.g.*, DDIM) is used to derive cleaner data at each timestep. During the sampling process, random noise $\tilde{\epsilon}_t$ is added to the intermediate sample, with its magnitude controlled by a sampling variance $\sigma_t$ in Eqn. (2). In the context of dual-branch editing paradigm, this added random noise progressively disturbs the structure of intermediate samples in the source branch, ensuring better editing capability to the target image to align with the target prompt. Typically, a larger magnitude of the added noise tends to reduce the structure consistency between the source and target images, but improves alignment with the target prompt.

Nevertheless, existing editing methods apply the same sampling variance to all pixels, either using a zero variance for structure preservation (Hertz et al., 2023), or a large non-zero variance to enhance editing capability (Huberman-Spiegelglas et al., 2024). Apparently, these methods ignore the varying degrees of structure adjustments required by different regions. We argue that regions corresponding to the essential semantics conveyed by the text prompt (*e.g.* the face of the cat shown in Figure 1) require more extensive edits, *i.e.*, structure adjustments. Inspired by this, we propose a spatially adaptive variance-guided sampling strategy, which works on adapting large sampling variance only to critical regions that are highly relevant to the target semantics for better editing capability.

Since the source and target branches share the same sampling strategy, we then introduce our general sampling strategy for them. We believe that critical regions typically represent the most essential parts of an object (*e.g.* the face of a cat) and are likely to be located inside the editing area. Therefore, we apply a large threshold $\delta'$ ($\delta' > \delta$) to the saliency map of the editing area $\boldsymbol{s}$, which is defined in Eqn. (10), to obtain the binary mask of those critical regions, denoted as $\boldsymbol{m}'$. $\boldsymbol{m}'_i = 1$ indicates that the $i$-th region of the image is a critical region. To enhance the editing capability for these regions, we apply a higher sampling variance to disturb their original structure, while the variance for other regions is set to 0 to maintain their structure. Formally, the $i$-th element of the variance matrix $\Sigma_t$ which is applied to the $i$-th pixel of the image is given by:

$$(\Sigma_t)_i = \begin{cases} \sigma_t & \text{if } \boldsymbol{m}'_i = 1 \\ 0 & \text{if } \boldsymbol{m}'_i = 0 \end{cases} \tag{11}$$

where $\sigma_t$ is defined in Eqn. (3) with a non-zero $\mu_t$. Then, for $t = T, \ldots, 1$, the sampling process defined in Eqn. (2) and Eqn. (3) is accordingly reformulated as:

$$\boldsymbol{z}_{t-1} = \sqrt{\alpha_{t-1}/\alpha_t}(\boldsymbol{z}_t - \sqrt{1 - \alpha_t}\epsilon_\theta(\boldsymbol{z}_t, t)) + \sqrt{1 - \alpha_{t-1} - \Sigma_t^2}\epsilon_\theta(\boldsymbol{z}_t, t) + \Sigma_t\tilde{\epsilon}_t, \tag{12}$$

where intermediate data $\boldsymbol{z}_t$ is replaced with $\boldsymbol{z}_t^{src}$ in the source branch, and $\boldsymbol{z}_t^{tgt}$ in the target branch. Notably, by simply replacing the variance term, our method can be easily applied to other sampling strategies, *e.g.*, DPM Solver++ (Lu et al., 2022b).

## 5 Experiments

### 5.1 Dataset and Evaluation Metrics

We follow recent works (Ju et al., 2024; Xu et al., 2024) to evaluate our method on the PIE-Bench (Ju et al., 2024), the only existing standardized benchmark for prompt-based image editing. In particular, PIE-Bench comprises 700 images with ten unique editing types. Each image is annotated with its source prompt, target prompt, blend words (*i.e.*, words specifying editing demands in the prompts), and the editing mask. Notably, only the source prompt, target prompt and blend words are needed in the task of prompt-based image editing, while the editing mask is used to evaluate the background preservation of the editing method. See Appendix A for more details of PIE-Bench.

For comprehensive evaluation, we follow Ju et al. (2024) to assess models on three aspects: a) structure consistency measured by distance between DINO self-similarities (Tumanyan et al., 2022), b) background preservation measured by PSNR, LPIPS (Zhang et al., 2018), MSE and SSIM (Wang et al., 2004), and c) target prompt-image alignment measured by CLIP similarity (Hessel et al., 2021). For detailed description of these metrics, please refer to Appendix B.

Table 1: Quantitative results of different methods.

| Method | | | Structure | Background Preservation | | | | CLIP Similarity | |
|---|---|---|---|---|---|---|---|---|---|
| Inverse | Sampling(steps) | Editing | Distance$^{\downarrow}_{\times 10^3}$ | PSNR$^{\uparrow}$ | LPIPS$^{\downarrow}_{\times 10^3}$ | MSE$^{\downarrow}_{\times 10^4}$ | SSIM$^{\uparrow}_{\times 10^2}$ | Whole$^{\uparrow}$ | Edited$^{\uparrow}$ |
| PnP-I | DDIM(50) | P2P-Zero | 51.13 | 21.23 | 143.87 | 135.00 | 77.23 | 23.36 | 21.03 |
| | | MasaCtrl | 24.47 | 22.78 | 87.38 | 79.91 | 81.36 | 24.42 | 21.38 |
| | | PnP | 24.29 | 22.64 | 106.06 | 80.45 | 79.68 | _25.41_ | **22.62** |
| | | P2P | **11.64** | _27.19_ | _54.44_ | _33.15_ | _84.71_ | 25.03 | 22.13 |
| | | **ViMAEdit** | _11.90_ | **28.75** | **43.07** | **28.85** | **85.95** | 25.43 | _22.40_ |
| EF | DPM-Solver++(20) | LEDITS++ | 23.15 | 24.67 | 80.79 | 118.56 | 81.55 | 25.01 | 22.09 |
| | | P2P | 14.52 | 27.05 | 50.72 | 37.48 | 84.97 | 25.36 | 22.43 |
| | | **ViMAEdit** | **14.16** | **28.12** | **45.62** | **33.56** | **85.61** | **25.51** | **22.56** |
| VI | DDIM(50) | P2P | 13.15 | 27.54 | 51.44 | 35.07 | 84.98 | 25.19 | 22.26 |
| | | **ViMAEdit** | **12.65** | **28.27** | **45.67** | **30.29** | **85.65** | **25.91** | **22.96** |
| VI | DDCM(12) | InfEdit | 13.78 | **28.51** | 47.58 | 32.09 | 85.66 | 25.03 | 22.22 |
| | DPM-Solver++(12) | **ViMAEdit** | **12.50** | 28.31 | **44.88** | **31.27** | **85.69** | **25.54** | **22.64** |

## 5.2 COMPARISON WITH EXISTING METHODS

We compare our method with seven diffusion model-based methods, *i.e.,* P2P-Zero (Parmar et al., 2023), MasaCtrl (Cao et al., 2023), PnP (Tumanyan et al., 2023), P2P (Hertz et al., 2023) with PnP Inversion (PnP-I) (Ju et al., 2024), P2P with Edit Friendly Inversion (EF) (Huberman-Spiegelglas et al., 2024), P2P with Virtual Inversion (VI) (Xu et al., 2024), and LEDITS++ (Brack et al., 2024), as well as one consistency model-based method, *i.e.,* InfEdit (Xu et al., 2024). For a comprehensive comparison, we apply all the three inversion techniques involved in baseline methods to our model, to obtain the initial noise for both branches. Meanwhile, we adopt the same sampling strategies, *i.e.,* DDIM (Song et al., 2020) and DPM-Solver++ (Lu et al., 2022b), as these methods except InfEdit. This is because InfEdit (Xu et al., 2024) introduces a new DDCM sampling tailored to the consistency model (Song et al., 2023; Luo et al., 2023) to achieve high generation quality in few-step settings. Since DDCM sampling is incompatible with the diffusion model, we adopt DPM-solver++ sampling, which also shows promising generation quality in few sampling steps, for comparing our model with InfEdit. Notably, for a fair comparison, we keep the number of sampling steps of our model as the same as InfEdit. Implementation details of our method are described in Appendix C.

We present the quantitative comparison results in Table 1. For a fair comparison, we unify the backbone of all diffusion model-based methods to be the pre-trained Stable Diffusion v1.5. For consistency model-based InfEdit, we keep its backbone (*i.e.,* the pre-trained LCM DreamShaper v7) unchanged. Overall, our ViMAEdit consistently outperforms previous works under different inversion and sampling settings, especially in terms of background preservation-related metrics and target image-prompt alignment-related metric (*i.e.,* CLIP Similarity). Notably, although it has been proved that consistency model can achieve higher generation quality in few denoising steps than diffusion models, our diffusion model-based ViMAEdit still outperforms consistency model-based InfEdit under the same sampling steps (*i.e.,* 12), demonstrating the effectiveness of our method.

In addition, we present qualitative comparison among different methods in Figure 4. Due to the limited space, for P2P, we only present its results of using the best-performing virtual inversion technique. As highlighted by red circles in the first example, baseline methods either alter the streetlight appearance or miss the fallen leaf in the source image, which are irrelevant to the editing intention. Similarly, in the second example, most baselines fail to preserve the boat on the sea or the stone to be the same as the source image. Although MasaCtrl preserves that, it fails to fulfill the intended editing, *i.e.,* changing the galaxy to sunset. In contrast, our method successfully edits the image according to the prompt with those elements in the background unchanged, demonstrating its better capability of the background preservation. Furthermore, in the third and the fourth examples, as compared to baseline methods, our method presents better alignment with the target prompt in converting the "snow" to "leaves" and changing the color of the flower, showcasing the superior editing capacity of our method. More qualitative results are provided in Appendix D.

## 5.3 ABLATION STUDY

To justify the effectiveness of our key designs, we introduce the following five model derivatives. 1) **w/o image embedding guidance**: we condition the noise prediction model solely on the text prompt. 2) **w/ CA-based grounding**: we use only cross-attention maps to ground the editing area. 3)

Table 2: Ablation study for our proposed three key designs.

| Method | Structure | Background Preservation | | | | CLIP Similarity | |
|---|---|---|---|---|---|---|---|
| | Distance$^{\downarrow}_{\times 10^3}$ | PSNR$^{\uparrow}$ | LPIPS$^{\downarrow}_{\times 10^3}$ | MSE$^{\downarrow}_{\times 10^4}$ | SSIM$^{\uparrow}_{\times 10^2}$ | Whole$^{\uparrow}$ | Edited$^{\uparrow}$ |
| **ViMAEdit** | 12.65 | 28.27 | **45.67** | **30.29** | **85.65** | 25.91 | 22.96 |
| w/o image embedding guidance | 12.91 | 27.47 | 51.58 | 34.89 | 84.89 | 25.40 | 22.55 |
| w/ CA-based grounding | 12.69 | 27.80 | 48.45 | 31.99 | 85.37 | 25.90 | 22.92 |
| w/ DPL-based grounding | 12.82 | 28.06 | 47.74 | 31.85 | 85.45 | 25.82 | 22.91 |
| w/ fixed sampling variance (zero) | **12.64** | **28.29** | 45.84 | 30.42 | 85.64 | 25.83 | 22.86 |
| w/ fixed sampling variance (non-zero) | 17.10 | 28.09 | 49.91 | 35.51 | 85.29 | **26.01** | **23.11** |

| Source image | Ours | InfEdit | P2P (VI) | LEDITS++ | PnP | MasaCtrl |
|---|---|---|---|---|---|---|

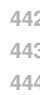

**Prompt:** a park bench and ~~red~~ pink trees in a flat style.

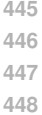

**Prompt:** the ~~galaxy~~ sunset over the durdle door.



**Prompt:** A ~~pink~~ blue flower with ~~yellow~~ red center in the middle.

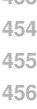

**Prompt:** a view of the mountains covered in ~~snow~~ leaves.

Figure 4: Qualitative results of different methods. We highlight incorrect editing parts by red circles.

**w/ DPL-based grounding**: we follow DPL to cluster self-attention maps for grounding the editing area. 4) **w/ fixed sampling variance (zero)**: we set the sampling variance to 0 for all pixels. 5) **w/ fixed sampling variance (non-zero)**: we set the sampling variance to $\sigma_t$ defined in Eqn. (3) for all pixels with $\mu_t = 1$. All the methods adopt the general DDIM sampling and virtual inversion.

From Table 2, we observe that by using image embeddings to guide the generation process, our method achieves significant improvement on background preservation and prompt alignment. This indicates that our target image embedding estimated from source image embedding benefits the source image background preserving and target semantics delivering. In addition, compared with CA-based grounding and DPL-based grounding, our editing area grounding strategy yields better performance on all evaluation aspects, while particularly promoting the background preservation. This suggests that the patch-to-patch relationships contained in self-attention maps do boost word-to-patch relationship refinement, and improve the editing area grounding. Last, we find that using the fixed zero sampling variance leads to slightly better structure consistency but worse background preservation and alignment with the target prompt, while using non-zero one leads to better alignment but significantly worse structure consistency and background preservation. Beyond them, our proposed spatially adaptive variance strikes a better balance on all the three evaluation aspects.

## 5.4 ANALYSIS AND DISCUSSION

**On estimated target image embeddings.** To gain more intuitive understanding of our estimated target image embeddings in capturing the target semantics, we condition the diffusion model solely on the estimated target image embedding with the target prompt and other operations (*i.e.*, inversion,

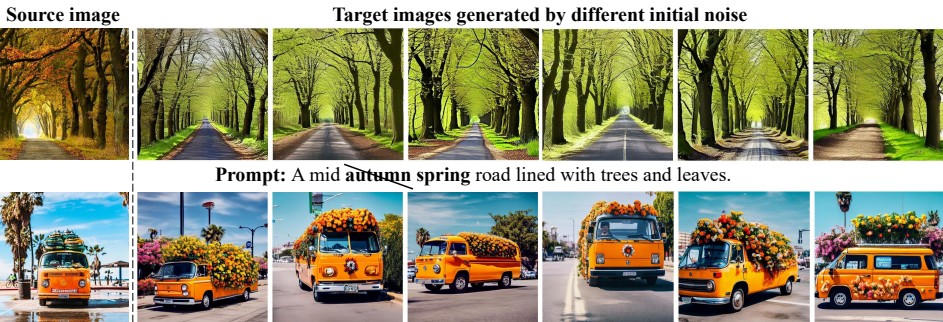

**Source image**    **Target images generated by different initial noise**

**Prompt:** A mid ~~autumn~~ **spring** road lined with trees and leaves.

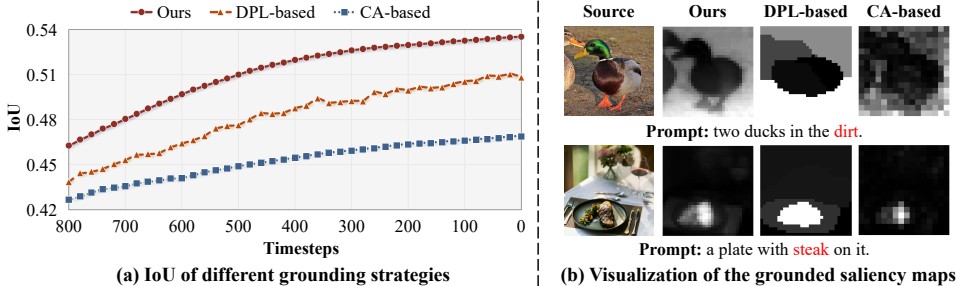

**Prompt:** An orange van with ~~surfboards~~ **flowers** on top.

Figure 5: Image editing solely based on the estimated target image embeddings without prompt.

(a) IoU of different grounding strategies

(b) Visualization of the grounded saliency maps

Figure 6: Comparison of different grounding strategies. Blend words are marked in red.

attention maps injection and editing area grounding) disabled. For comprehensive justification, we perform the generation process for six times, each with a random initial noise. Two examples are shown in Figure 5. As can be seen, in the first example, the road and the forest in the source image, as well as their layout, are well preserved in all generated images, while the season is all transformed from the "autumn" to the desired "spring". In the second example, each of the generated images contains a van similar to the source image, except that the "surfboards" on the roof is replaced with "flowers". Both examples demonstrate that our estimated target image embeddings capture the desired semantics well, which not only preserve the essential visual contents in the source image that should be maintained in target images, but also align well with the target prompts.

**On editing area grounding.** To gain deeper understanding of our proposed editing area grounding strategy, we compare the grounding precision of our method with aforementioned two derivatives: w/ CA-based grounding and w/ DPL-based grounding. Specifically, we compute the average intersection over union (IoU) between the grounded mask and the groundtruth editing mask. As shown in Figure 6a, our method consistently achieves more precise grounding results at all timesteps, demonstrating the effectiveness of iteratively using patch-to-patch relationships captured by self-attention maps for grounding result refinement. In Figure 6b, we further visualize the saliency maps obtained from different methods. As we can see, compared with the other derivatives, our method highlights the regions that need to be changed more precisely. Both the quantitative and qualitative results show the effectiveness of our proposed editing area grounding strategy.

## 6 CONCLUSION

In this work, we propose a vision-guided and mask-enhanced adaptive editing (ViMAEdit) method for prompt-based image editing. Based on the commonly used dual-branch editing paradigm, we introduce an image embedding-enhanced denoising process, a self-attention-guided iterative editing area grounding strategy, and a spatially adaptive variance-guided sampling strategy, to promote the target image generation. Extensive experiments demonstrate the superiority editing capability of our method over all existing methods. In particular, experiment results verify the effectiveness of each key design in our proposed ViMAEdit, confirming the benefits of directly introducing the image embedding as additional guidance in the denoising process, fulfilling editing area grounding by leveraging patch-to-patch relationships to refine word-to-patch relationships, and adopting spatially adaptive variance for sampling.

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

## A  PIE-BENCH

PIE-Bench (Ju et al., 2023; 2024) comprises 700 images in natural and artificial scenes (*e.g.*, paintings) with ten unique editing types: (0) random editing written by volunteers, (1) changing object, (2) adding object, (3) deleting object, (4) changing object content, (5) changing object pose, (6) changing object color, (7) changing object material, (8) changing background, and (9) changing image style. For each editing type (except type 0), the numbers of images belonging to the two scenes (*i.e.*, natural and artificial scenes) are the same. Within each scene, images are evenly distributed among four categories: animal, human, indoor environment, and outdoor environment.

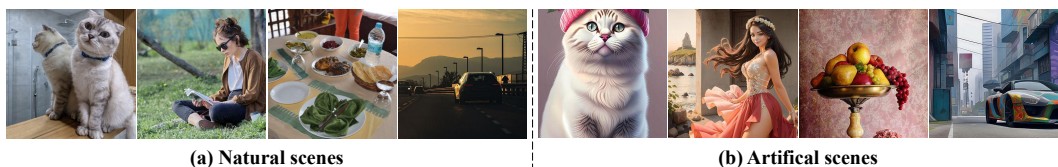

(a) Natural scenes                    (b) Artifical scenes

Figure 7: Examples in PIE-Bench among natural and artificial scenes.

## B  EVALUATION METRICS

We assess our method on the following three aspects.

- **Structure Consistency**. Following Tumanyan et al. (2022), we first leverage the self-similarity of spatial features extracted from DINO (Caron et al., 2021) as the structure representation, which is proven to capture the image structure while ignoring the appearance information. Then, we compute the distance between structure representations of two images using MSE to evaluate their structure consistency.

- **Background Preservation**. Following Ju et al. (2023), we regard the areas outside the manual-annotated editing mask provided by PIE-bench as the background for each image. We calculate standard PSNR, LPIPS (Zhang et al., 2018), MSE, and SSIM (Wang et al., 2004) between the background areas of the target image and source image to assess the background preservation.

- **Target Prompt-Image Alignment**. Following Hessel et al. (2021), we employ CLIP(Radford et al., 2021) to calculate the similarity between the target prompt embedding and the target image embedding. Specifically, following Ju et al. (2023), we use both the whole target image and the editing area of the target image (with all pixels outside the editing mask blacked out) for computing the CLIP similarity. These two results are denoted as Whole Image CLIP Similarity and Edit Region CLIP Similarity, respectively.

## C  IMPLEMENTATION DETAILS

During the image embedding-enhanced denoising process, the multiplier $\lambda$ of the guided image embedding is set to $0.4$. Regarding the self-attention-guided iterative editing area grounding, we set the mask threshold $\delta$, the iteration weight $\gamma$, iteration number $N$ to $0.5$, $0.5$ and $5$, respectively. Pertaining to the spatially adaptive variance-guided sampling, we set the mask threshold $\delta'$ for localizing critical regions to $0.8$, and the magnitude of the variance $\mu_t = 1$. All experiments are conducted on an A100 GPU with PyTorch.

## D  MORE QUALITATIVE RESULTS

We present more qualitative results below to comprehensively demonstrate the superior editing capacity of our method among both natural and artificial scenes.

## D.1 NATURAL SCENES

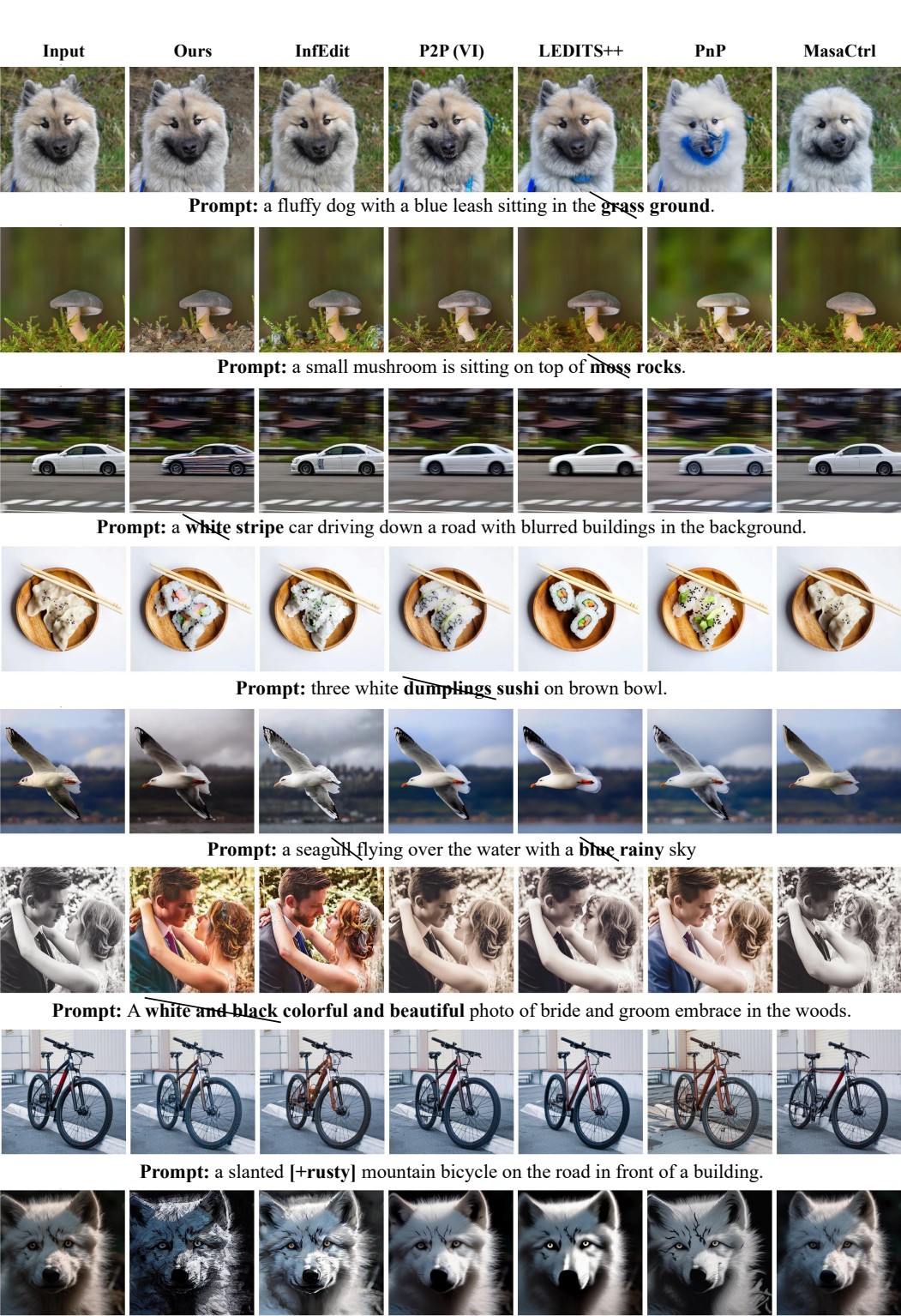

Figure 8: More qualitative results.

## D.2 ARTIFICIAL SCENES

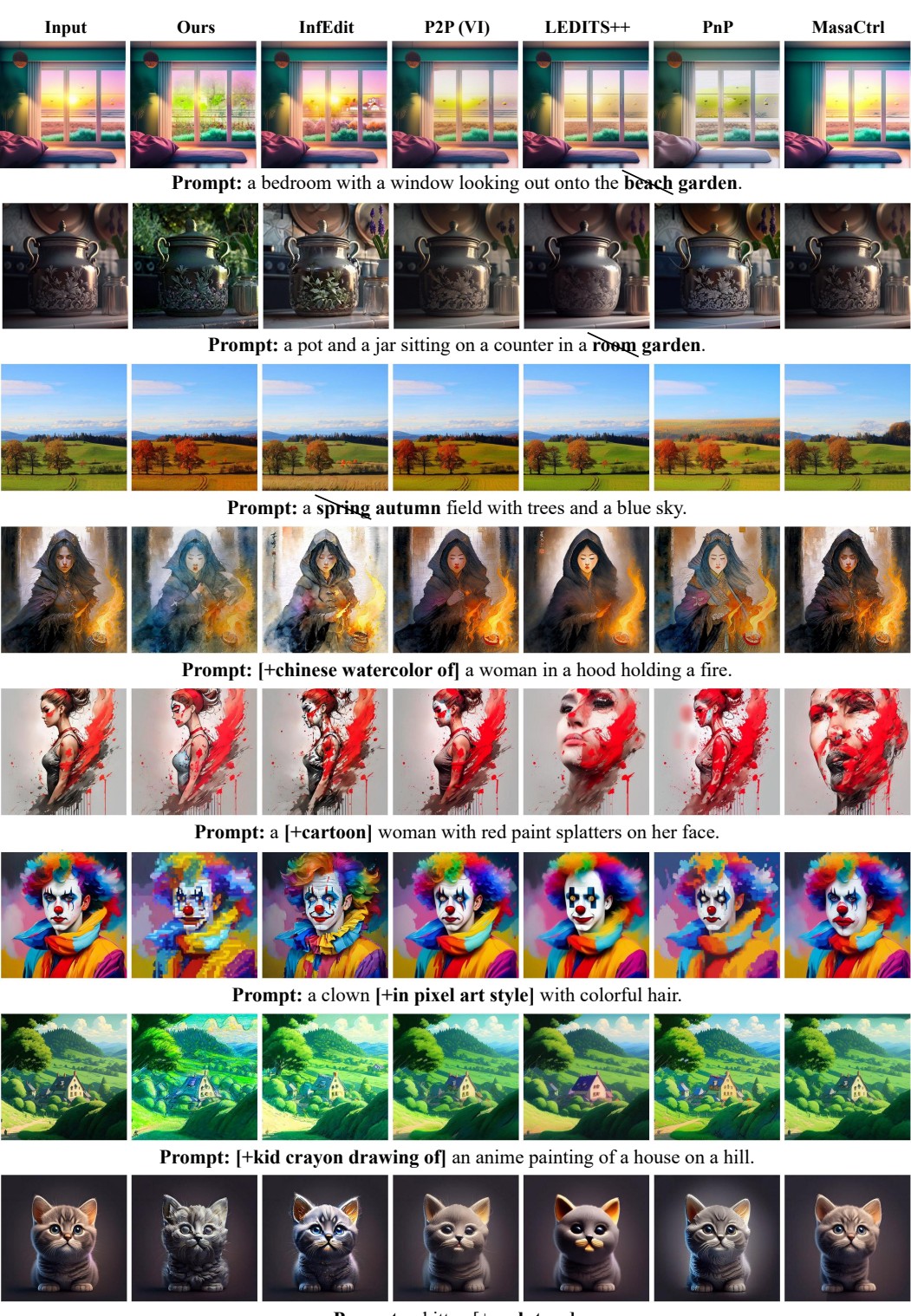

Figure 9: More qualitative results.

