# OpenReview forum: "Vision-guided and Mask-enhanced Adaptive Denoising for Prompt-based Image Editing"
_ICLR.cc/2025/Conference — ICLR 2025 Conference Withdrawn Submission_

### Official Review · Reviewer_oqBD · 2024-10-23

**Soundness:** 3
**Presentation:** 3
**Contribution:** 2
**Rating:** 6
**Confidence:** 4

**Summary:**

This paper proposes a Vision-guided and Mask-enhanced Adaptive Editing (ViMAEdit) method for prompt-based image editing, which shows promising results in addressing the limitations of existing methods. Specifically, the ViMAEdit method incorporates three key novel designs. First, it uses image embeddings for enhanced conventional textual prompt-based denoising process. Second, it has a self-attention-guided iterative editing area grounding strategy. Last, it presents a spatially adaptive variance-guided sampling. Experimental results demonstrate the superior editing capacity of ViMAEdit over all existing methods.

**Strengths:**

1. The paper is well-written, with a clear and logical structure. The ideas are presented in an organized manner. The attached related code is a significant pros as it allows for reproducibility and further exploration of the proposed method.

2. The performance of the proposed method appears to be pretty good and convincing. The method shows superiority over existing methods in multiple aspects such as structure consistency, background preservation, and target prompt - image alignment. These results suggest that the ViMAEdit method has the potential to make a substantial impact in the field of prompt-based image editing and could serve as a benchmark for future research in this area.

**Weaknesses:**

1. Lack the comprehensive comparison with a wide range of image editing methods. i.e., [1][2][3]



[1] InstructPix2Pix: Learning to Follow Image Editing Instructions

[2] MagicBrush: A Manually Annotated Dataset for Instruction-Guided Image Editing

[3] Guiding Instruction-based Image Editing via Multimodal Large Language Models

**Questions:**

Refer to weaknesses.

---

### Official Review · Reviewer_PUJc · 2024-10-29

**Soundness:** 2
**Presentation:** 3
**Contribution:** 2
**Rating:** 1
**Confidence:** 5

**Summary:**

The paper introduces ViMAEdit, a method for prompt-based image editing that uses image embeddings to enhance guidance in generating target images. To improve the precision of edits, the method employs a self-attention-guided iterative strategy, refining word-to-patch relationships by utilizing patch-to-patch connections from self-attention maps. Additionally, the paper proposes a spatially adaptive variance-guided sampling method that prioritizes significant regions in the image, enhancing the overall quality and accuracy of edits.

**Strengths:**

- The paper is clearly written and accessible, allowing readers to grasp the methodology and contributions effectively.
- Qualitative results in the main text and supplementary materials demonstrate high performance, with visually compelling results across various editing tasks.

**Weaknesses:**

1. **Lack of Clarity on Prompt Limitations**: The authors claim in Line 66 that "the target prompt can only highlight the core editing intention... and its capacity to depict finer visual details... is inadequate." However, this statement appears unsupported, as well-designed prompts can indeed capture nuanced visual details, such as color or expression changes in the edited subject. This limitation is not sufficiently justified or empirically verified.

2. **Justification for Method Choice**: While image embeddings are presented as providing more accurate guidance, it is unclear what specific challenges led to the development of this approach over existing methods. An explicit discussion on the limitations of prior approaches and the specific advantages offered by ViMAEdit’s embeddings would provide valuable context.

3. **Contradictory Assertions on Attention-Based Methods**: The claim in Line 74 that "existing methods mostly only rely on cross-attention maps...to locate the editing area as patches relevant to the blend words" seems inaccurate, as methods such as MasaCtrl allow both auto-aggregated masks and user-specified masks, while others like PnP do not require a mask, and LEDITS++ employs averaged masks for all editing tokens. This generalization appears to overlook these alternatives.

4. **Potential for Misidentifying Editing Areas**: The self-attention-guided iterative grounding module aims to refine word-to-patch relationships to produce a mask targeting the salient object. However, this approach may struggle when the target area is not the most prominent object, such as when editing a minor object in a detailed scene. It remains unclear how the proposed method manages cases where the target object is not the most visually dominant element.

5. **Sampling Variance and Editing Strength**: The assertion in Line 87 that "sampling variance controls the editing strength applied to each pixel..." seems overstated. While increased sampling variance introduces greater variation, this does not necessarily translate to stronger or more controlled editing. This approach risks introducing inconsistencies without a clear mechanism for managing variance across different parts of the image.

6. **Inconsistencies with Dissimilar Prompts**: The paper proposes an image embedding-enhanced denoising process, estimating the target image embedding based on the source image using Equation (7). If there is a significant semantic gap between the source and target prompts, however, the resulting embedding may lack coherence or relevance. How the method adapts to or mitigates inconsistencies arising from such disparities is not addressed.

7. **Excessive Alterations in Results**: Figure 5 shows cases where the method alters background details or adjusts object pose and view significantly. This behavior may indicate a lack of control over the degree and specificity of the edits, suggesting further refinement is necessary for tasks requiring subtler changes.

8. **Absence of User Study for Evaluation**: A user study evaluating the method's efficacy in real-world applications, such as its performance on varied and user-specified edits, would add significant value. User studies are instrumental in assessing how well the method aligns with user intent and the overall editing experience.

9. **Complexity in Practical Applications**: Real-world image editing tasks often involve scenes with multiple objects, where edits must be made based on flexible prompts. The need to specify a source prompt, blend word, and various conditions can complicate the process and may restrict the practical usability of the system for typical users seeking simpler image adjustments.

**Questions:**

1. **Handling Object Removal and Pose Adjustments**: How does the proposed method handle object removal, significant pose alterations, or other extensive adjustments? Additional visual examples demonstrating these capabilities would be helpful.

2. **Parameter Optimization and Comparative Evaluation**: Given that methods such as MasaCtrl, LEDITS++, and PnP have diverse parameter configurations, has ViMAEdit been evaluated against these methods under varied parameter settings? Further information on how parameters were optimized for the specific dataset would clarify the robustness of comparisons.

3. **Comparison with Alternative Methods**: Why are DiffEdit [1] and FPE [2] not included in the comparison? Including these could provide a broader perspective on ViMAEdit’s relative performance.

4. **Absence of a Discussion on Limitations**: Acknowledging the limitations of ViMAEdit would provide a balanced view and inform future work on areas needing improvement. What are the primary constraints of this method, and how might these affect its applicability?
#### References
1. Diffedit: Diffusion-based semantic image editing with mask guidance. *International Conference in Learning Representations*, 2023.

2. Towards understanding cross and self-attention in stable diffusion for text-guided image editing. In *Proceedings of the IEEE/CVF Conference on Computer Vision and Pattern Recognition*, pp. 7817–7826, 2024.

---

### Official Review · Reviewer_WBC4 · 2024-10-29

**Soundness:** 3
**Presentation:** 3
**Contribution:** 3
**Rating:** 6
**Confidence:** 3

**Summary:**

This paper presents a method called Vision-guided and Mask-enhanced Adaptive Editing (ViMAEdit) for prompt-based image editing. The introduced method combines three key components: CLIP-based target image embedding estimation, a self-attention-guided iterative editing area grounding strategy, and spatially adaptive variance-guided sampling. Experimental results indicate that this method improves the quality and precision of prompt-based image editing, offering superior editing capabilities compared to existing methods.

**Strengths:**

- The paper introduces a well-designed method that effectively addresses the limitations of prompt-based text-to-image diffusion models by leveraging vision-guided and mask-enhanced adaptive editing (ViMAEdit) for more precise control over generated images.

- The use of  iterative editing area grounding and spatially adaptive variance-guided sampling are innovative approaches that contribute significantly to improving image quality and control in the editing process.

- The writing and presentation are generally clear and accessible, making it easier for readers to follow the proposed method.

**Weaknesses:**

- The paper introduces $\triangle e$ to represent the difference between text embeddings in the CLIP embedding space. However, it is unclear how  $\triangle e$ and the image embeddings are processed together. Specifically, for models like Stable Diffusion 1.5, the text embedding dimension is $77 \times 768$. How does the model align this dimension with the image embeddings, which typically have a different structure? More clarification on the dimensional operations and how the embeddings are effectively integrated is required.

- The proposed approach utilizes an IP-Adapter module, which is trained separately and introduces additional knowledge. This creates an unfair advantage compared to other baseline methods that do not use such a pre-trained module. It would be more appropriate to conduct separate ablation studies to highlight the impact of IP-Adapter or to compare methods without the additional pre-trained components to ensure fair comparisons.

- Typo error: On line 309, "$ P_2 \times P_2$" should be "$ P_2$ ."

**Questions:**

Please refer to the Weaknesses for more details.

---

### Official Review · Reviewer_4fZE · 2024-11-03

**Soundness:** 3
**Presentation:** 3
**Contribution:** 2
**Rating:** 5
**Confidence:** 5

**Summary:**

The paper proposes a novel method called Vision-guided and Mask-enhanced Adaptive Editing (ViMAEdit), which combines three innovative components: CLIP-based target image embedding, self-attention-guided iterative editing area grounding, and spatially adaptive variance-guided sampling. These components significantly enhance the control and accuracy of the image editing process.

**Strengths:**

1.  The paper proposes a novel method called Vision-guided and Mask-enhanced Adaptive Editing (ViMAEdit), which combines three innovative components: CLIP-based target image embedding, self-attention-guided iterative editing area grounding, and spatially adaptive variance-guided sampling. These components significantly enhance the control and accuracy of the image editing process.

2. Enhanced Editing Performance: By leveraging image embeddings as explicit guidance, ViMAEdit improves the effectiveness of prompt-based image editing, addressing the limitations of conventional text-only prompt-based methods. The use of iterative patch-to-patch refinement also allows for more precise region targeting during editing.

3. The experimental evaluation shows that ViMAEdit outperforms existing methods in terms of editing quality and adaptability.

**Weaknesses:**

1. Comparison with State-of-the-Art Methods:
It would be more validating if the proposed method were compared with state-of-the-art methods, such as Guide-and-Rescale [1] and MAG-Edit [2]. Especially, MAG-Edit, which has a very similar setting.

2. Ablation Study Lacks Qualitative Results: The ablation study lacks qualitative results, which are essential to confirm the role of the image embedding mentioned in lines 65-72. Including qualitative visualizations would help clarify the importance of each component in the proposed approach.

3. Insufficient Explanation of Why Proposed Key Aspects matter: The qualitative experimental results show the advantage of the proposed method in the alignment of adjectives such as color editing, but it is not clear which specific techniques contribute to achieving good alignment. From our experience, aligning adjectives like in Figure 4 (the example of the flower) is challenging if P2P can not address it well. The paper could benefit from explaining why the adjective alignment works effectively and which methods facilitate this, such as in MAG-Edit[2], which proposes cross-attention alignment. Since a better grounding technique could potentially preserve the background more effectively without negatively impacting the editing results, it could not contribute to the alignment of editing regions. Authors are suggested to explain why their method can do well in such a complicated setting like figure 4 a blue flower with red in the center.

4. Single Object Limitation: The results presented in the paper only contain a single object. It is unclear if the self-attention guided grounding can effectively locate a single object in images with multiple objects (for instance, multiple dogs, only edit one dog). This limitation should be addressed to demonstrate the method's applicability in more complex scenarios. Although the self-attention-guided iterative region grounding is a promising approach, its robustness in challenging or diverse image scenarios is not sufficiently addressed. It would be beneficial for the authors to include additional experiments to verify the method's consistency across varied and complex datasets.

5. Time-Consumption of Iterative Grounding: There is no discussion on whether the iterative grounding process is time-consuming. The authors should provide an analysis of the computational cost of the iterative approach.

6. User Study: Including a user study would enhance the credibility of the proposed method.


References:
[1] Titov V, Khalmatova M, Ivanova A, et al. Guide-and-Rescale: Self-Guidance Mechanism for Effective Tuning-Free Real Image Editing[C]//European Conference on Computer Vision. 2024.
[2] Mao Q, Chen L, Gu Y C, Fang Z, Shou M Z. MAG-Edit: Localized Image Editing in Complex Scenarios via Mask-Based Attention-Adjusted Guidance[C]//Proceedings of the 32nd ACM International Conference on Multimedia. 2024: 6842-6850.

**Questions:**

Please see the weaknesses. Authors are well encouraged to compare the proposed method with state-of-the-art techniques, include qualitative results in the ablation study, and clarify which techniques contribute to the advantages of their methods in attribute editing.

---

### Note · Authors · 2024-11-13

I have read and agree with the venue's withdrawal policy on behalf of myself and my co-authors.